# Zebrafish Model as a Screen to Prevent Cyst Inflation in Autosomal Dominant Polycystic Kidney Disease

**DOI:** 10.3390/ijms22169013

**Published:** 2021-08-20

**Authors:** Inês Oliveira, Raquel Jacinto, Sara Pestana, Fernando Nolasco, Joaquim Calado, Susana Santos Lopes, Mónica Roxo-Rosa

**Affiliations:** 1CEDOC, Chronic Diseases Research Center, NOVA Medical School|Faculdade de Ciências Médicas, Universidade Nova de Lisboa, Campo Mártires da Pátria, 130, 1169-056 Lisboa, Portugal; imoliveira93@gmail.com (I.O.); jacinto.a.r@gmail.com (R.J.); sara.pestana@nms.unl.pt (S.P.); 2Department of Nephrology, Centro Hospitalar e Universitário de Lisboa Central, Hospital de Curry Cabral, Rua da Beneficência, 8, 1069-166 Lisboa, Portugal; febnolasco@gmail.com (F.N.); jcalado@nms.unl.pt (J.C.); 3ToxOmics, Center of ToxicoGenomics & Human Health, NOVA Medical School|Faculdade de Ciências Médicas, Universidade Nova de Lisboa, Campo Mártires da Pátria, 130, 1169-056 Lisboa, Portugal

**Keywords:** autosomal dominant polycystic kidney disease (ADPKD), cystic fibrosis transmembrane conductance regulator (CFTR), Kupffer’s vesicle (KV), polycystin-2 (PC2)

## Abstract

In autosomal dominant polycystic kidney disease (ADPKD), kidney cyst growth requires the recruitment of CFTR (cystic fibrosis transmembrane conductance regulator), the chloride channel that is defective in cystic fibrosis. We have been studying cyst inflation using the zebrafish Kupffer’s vesicle (KV) as model system because we previously demonstrated that knocking down polycystin 2 (PC2) induced a CFTR-mediated enlargement of the organ. We have now quantified the PC2 knockdown by showing that it causes a 73% reduction in the number of KV cilia expressing PC2. According to the literature, this is an essential event in kidney cystogenesis in ADPKD mice. Additionally, we demonstrated that the PC2 knockdown leads to a significant accumulation of CFTR-GFP at the apical region of the KV cells. Furthermore, we determined that KV enlargement is rescued by the injection of Xenopus *pkd2* mRNA and by 100 µM tolvaptan treatment, the unique and approved pharmacologic approach for ADPKD management. We expected vasopressin V2 receptor antagonist to lower the cAMP levels of KV-lining cells and, thus, to inactivate CFTR. These findings further support the use of the KV as an in vivo model for screening compounds that may prevent cyst enlargement in this ciliopathy, through CFTR inhibition.

## 1. Introduction

Autosomal dominant polycystic kidney disease (ADPKD) is a leading cause of end-stage kidney disease (ESKD). In 1957, Dalgaard et al. estimated 1 in 400–1000 newborns worldwide to be at risk of being affected with ADPKD over an 80-year lifetime [1]. Supporting this rate, recent population-based whole-genome sequencing data showed that protein-truncating and clinically confirmed mutations provide a lifetime risk of ADPKD of at least 9 cases per 10,000 individuals [2].

ADPKD is a ciliopathy induced by mutations in the *PKD1* (OMIM-601313) or *PKD2* (OMIM-613095) genes, in 72–75% and 15–18% of families, respectively [3]. These two genes encode the mechanosensor polycystin-1 (PC1) and the calcium-permeable non-selective cation channel polycystin-2 (PC2), respectively [4,5]. These two transmembrane proteins are crucial in regulating intracellular calcium homeostasis in the kidney epithelium [4,5]. Indeed, the loss or dysfunction of either PC1 or PC2 leads to a reduction in basal intracellular calcium levels, which is thought to trigger cystogenesis [4,5]. Whether this regulation starts with a ciliary calcium wave mediated by the PC1–PC2 complex is under debate [6]. Nevertheless, as recently demonstrated, the ciliary expression of PC2 is essential to prevent kidney cystogenesis in an ADPKD mouse model [7]. Mutant mice carrying a non-ciliary localized but fully functional PC2 still develop embryonic kidney cysts that appear indistinguishable from those of mice completely lacking PC2 [7]. PC2 is also thought to work together with IP3 receptors in order to regulate intracellular calcium release from the endoplasmic reticulum pool [8].

Inflation and continuous expansion of the cysts is assured by an transepithelial fluid secretion into the lumen [9]. The countless fluid-filled cysts destroy kidney function. About half of the patients develop ESKD, requiring dialysis and kidney transplantation by age 60. The cystic fibrosis transmembrane conductance regulator (CFTR, OMIM 602421) is a key player in the cyst inflation process, being expressed and activated in ADPKD cyst-lining cells [10,11,12,13,14]. CFTR is essential in the regulation of ion and fluid transport in epithelia and its dysfunction causes cystic fibrosis [15]. Supporting the involvement of CFTR in ADPKD cyst inflation, it was shown that fluid accumulation within cysts involves CFTR-like chloride currents [14] and it is slowed down either through inhibition or knockdown of CFTR [10,11,12,13,14]. Interestingly, besides the pseudo-Bartter syndrome, a transient hyponatremic, hypochloremic metabolic alkalosis observed in some cystic fibrosis infants, that has been mainly attributed to the excessive loss of NaCl in the sweat [16], no other functional/structural kidney phenotype has been reported in cystic fibrosis patients. And, in fact, a milder kidney phenotype was observed in patients affected by both ADPKD and cystic fibrosis [9]. The in vivo mechanisms involved in the activation of CFTR during kidney cyst inflation are still emerging.

CFTR activation requires its prior cAMP-dependent phosphorylation by PKA [17,18,19,20]. Therefore, as the lack of calcium homeostasis raises the intracellular levels of cAMP in ADPKD cells, CFTR has been pointed to as a downstream effector of cAMP in cyst growth [9]. This has led to several studies testing drugs targeting kidney cAMP production, tolvaptan being the most promising. Acting as a vasopressin V2 receptor (V2R) antagonist, tolvaptan lowers the intracellular cAMP levels of cyst-lining cells. It is effective in slowing down the CFTR-mediated enlargement of the cysts and in decelerating the disease progression. Therefore, tolvaptan was recently approved in Japan, Canada, EU, and USA for the management of ADPKD (TEMPO 3:4 clinical trial) [21]. However, it leads to heavy aquaretic side effects which are relevant when considering tolvaptan as a life-long treatment [21]. This limits the number of patients who are eligible for treatment [22], highlighting the need for other therapeutic targets. In this context, we consider that CFTR trafficking towards the apical membrane of the cell, and its stability once there, must be investigated in ADPKD tissues. Indeed, it is well established that these are relevant features in controlling the number of active CFTR molecules at the membrane and thus the overall chloride transport [15]. Although few, there are relevant reports indicating that overexpression of PC1 decreases apical expression of CFTR [23] and showing that cyst-lining cells from ADPKD patients do express CFTR apically [14].

Several human and mouse kidney-derived cellular models have been used for drug screening for ADPKD [10,11,24,25]. Despite their undoubted importance, they lack the organ-to-organ communication and the cellular and molecular environments offered by an animal model. Some authors have used magnetic resonance imaging to follow the in vivo effect of pharmacological approaches on cyst growth in ADPKD mouse models [26] However, a simpler and less expensive model is needed in high throughput screening assays. Therefore, although it is not a kidney-related organ, we have proposed in the past the zebrafish Kupffer’s vesicle (KV) as a model organ for the cyst inflation process [27]. This organ is present in the early embryonic development of zebrafish, in a time window of seven hours that runs from the somite stages (ss) 1 to 14. KV is a fluid-filled vesicle whose inflation depends on CFTR activity [27,28]. The epithelial cells lining the KV each have one cilium and 80% of KV cilia become motile by 10 ss [29]. Cilium motility is essential for the generation of a characteristic fluid flow required for the early establishment of the left–right laterality of the internal organs of the fish [29,30,31,32,33]. Relevant to this study is that, mirroring the ADPKD cyst inflation process, *pkd2* knockdown causes a significant increase in CFTR-mediated fluid secretion into the KV lumen [27]. We demonstrated that such KV enlargement can be rescued by the injection of *Xenopus pkd2*-mRNA. Moreover, we explored whether such KV enlargement is accompanied by changes in CFTR expression. We concluded that PC2 knockdown led to an accumulation of CFTR, namely at the apical region of the KV cells. Finally, knowing that KV epithelial cells express *vasopressin receptor 2-like* gene, we tested the ability of tolvaptan to rescue the KV enlargement of the PC2-knockdown embryos. Our positive results reinforce the utility of this ADPKD-cyst-model organ to test the potential of other molecules interfering with CFTR trafficking, stability and activity in preventing ADPKD cyst growth.

## 2. Results and Discussion

### 2.1. PC2 Knockdown

As previously demonstrated, the one-cell stage injection of the anti-*pkd2* mRNA translational blocking morpholino (*pkd2*-augMO) used in our studies is efficient in knocking down the PC2 in the KV time window [27]. To quantify this efficiency, we compared the percentage of cilia showing ciliary staining of PC2 between wild type (WT) and *pkd2*-augMO embryos. Ciliary staining of PC2 was detected in about 79 ± 13% and 76 ± 17% of the KV cilia of WT and mismatch-MO- (*pkd2* control morpholino) injected embryos, respectively (Figure 1). In contrast, a faint PC2 signal was detected along the ciliary membrane in only 21 ± 10% of the KV cilia of *pkd2*-augMO embryos (Figure 1). Thus, the one-cell stage injection of the *pkd2*-augMO caused a 73% reduction (*p* < 0.0001) of the KV cilia expressing PC2. These findings further support our recently published data showing that PC2 knockdown resulted in shorter KV cilia length and abnormal KV flow dynamics [34]. Framing these results with the recent findings of Walker et al. who showed that the ciliary exclusion of PC2 is an essential event for kidney cystogenesis in ADPKD mice [7], the results strengthen the use of the *pkd2*-augMO to study the impact of the lack of PC2 on the KV. This same *pkd2*-augMO has been used and validated by other groups [35,36]. On the other hand, the zebrafish mutant for the orthologous gene of human *PKD2*, the *curly-up* (*cup*^−/−^) mutant, is inadequate to accurately study the impact of knocking down *pkd2* in the KV. Indeed, *cup^−/−^* mutant embryos have a maternal contribution of *pkd2* mRNA from their heterozygotic mother. PC2 protein is, therefore, present in the KV time window in the *cup^−/−^* mutant embryos, being still detected in pronephric cilia, much later in their development (at 36 h post fertilization, hpf) [27,36].

### 2.2. Knockdown of PC2 Leads to the Apical Accumulation of CFTR Protein

The literature recognizes CFTR as a key player in ADPKD cyst inflation [9,10,11,12,13,14]. Whether it results from enhanced CFTR activity alone or whether it also requires higher amounts of CFTR protein at the apical cell membrane, it is not known. Our attempts to quantify the CFTR protein levels, either by Western-blot or by whole-mount immunostaining failed as none of the tested commercially available antibodies against human CFTR cross-reacted with the zebrafish protein. Therefore, to tackle this important question, we evaluated the KV CFTR-GFP fused protein expression in *TgBAC(cftr-GFP)pd1041* zebrafish, upon the knockdown of *pkd2*. In agreement with the findings of Navis et al. [28], our experiments at early developmental stages (from 2 to 14 ss) showed that CFTR-GFP expression was restricted to KV-lining cells (Figure 2), namely at the apical region of these cells (Figure 3A–C).

We then compared the mean fluorescence intensity (MFI) of the CFTR-GFP signal, throughout the KV 3D structure, of 8–10 ss *pkd2*-knockdown embryos and non-injected and mismatch-MO injected controls. We determined the MFI of the KV images resulting from the sum of all slices of each KV scan of 15 WT, 6 mismatch-MO, and 20 *pkd2*-augMO injected embryos (Figure 3). WT and mismatch-MO injected embryos showed no statistically significant difference in their normalized MFIs (Figure 3D,E,M). Contrastingly, *pkd2*-knockdown KVs showed 1.9 times higher normalized MFIs than the non-injected siblings (*pkd2*-knockdown normalized MFI = 2.8 × 10^6^ ± 1.4 × 10^6^ vs. non-injected control normalized MFI = 1.5 × 10^6^ ± 1.0 × 10^6^, with *p* = 0.0024) (Figure 3D,F,M), suggesting higher CFTR-GFP protein levels. To confirm our results by another approach, we additionally compared the CFTR-GFP-MFIs of *pkd2*-knockdown embryos to those of non-injected siblings by flow cytometry (BD FACS-Canto II, BD Biosciences, Franklin Lakes, NJ, USA) (Figure 4). In comparison to the auto-fluorescence of WT embryos (AB control) (Figure 4A), a GFP-positive cell population was clearly detected in *TgBAC(cftr-GFP)pd1041* samples (red arrow in Figure 4B) which corresponded to KV cells. Corroborating the results of the confocal live-scan analysis, for an equal number of GFP-positive cells (Figure 4E), *pkd2*-kockdown embryos presented on average 1.4 times higher CFTR-GFP normalized MFI than their non-injected siblings (*pkd2*-knockdown normalized MFI = 10.95 ± 1.35 vs. non-injected control normalized MFI = 7.80 ± 1.94, with *p* = 0.0087) (Figure 4F). Interestingly, a KV microarray analysis in our lab (unpublished data) revealed that PC2 knockdown had no effect on *cftr* transcriptional levels (Appendix A). Therefore, we did not expect to have more CFTR-GFP synthesis, leading us to hypothesize that the higher amounts of CFTR-GFP observed in *pkd2*-knockdown *TgBAC(cftr-GFP)pd1041* KV cells resulted from protein stabilization.

As already mentioned, the *TgBAC(cftr-GFP)pd1041* whole KV live-scans clearly presented an accumulation of CFTR-GFP at the apical region of the KV cells (highlighted in red in Figure 3G). This region should include CFTR-GFP molecules inserted in the apical membrane as well as those located in sub-apical vesicles that control the membrane stability of the protein. These are vesicles involved in the anterograde trafficking, endocytosis, and recycling of the protein [37]. When focusing on these regions specifically, we concluded that CFTR-GFP accumulated 1.4 times more at the apical region of the KV cells of *pkd2*-knockdown embryos (*pkd2*-augMO apical-MFI = 3.9 × 10^5^ ± 2.1 × 10^5^ vs. non-injected control apical-MFI = 2.7 × 10^5^ ± 1.4 × 10^5^, with *p* = 0.0422) (Figure 3G,I,M), suggesting enhanced membrane stability of the protein. No significant difference was observed when comparing the apical MFI of non-injected embryos with that of mismatch-MO control (mismatch-MO apical-MFI = 2.5 × 10^5^ ± 1.4 × 10^5^ vs. non-injected control apical-MFI = 2.7 × 10^5^ ± 1.4 × 10^5^) (Figure 3G,H,M). Although informative about the effect of the absence of PC2 over the CFTR-GFP expression, we cannot exclude that CFTR-GFP turnover may differ from that of native CFTR. Therefore, it will be interesting to validate these results, evaluating the native CFTR using in vitro cellular models, in the future.

As expected, the *pkd2*-knockdown translated into an increase of the KVs volume, corroborating our previous findings [27]. Indeed, on average *pkd2*-knockdown *TgBAC(cftr-GFP)pd1041* embryos with 8–10 ss, presented KVs with 1.5 times the volume of non-injected siblings (*pkd2*-knockdown KV_volume_ = 133 × 10^3^ ± 37 × 10^3^ μm^3^ vs. non-injected control KV_volume_ = 88 × 10^3^ ± 34 × 10^3^ μm^3^, with *p* = 0.0061) (Appendix A).

In conclusion, our data showed that reduced levels of PC2 enhanced the CFTR apical localization, probably through protein stabilization, likely contributing to the CFTR-mediated KV enlargement of the *pkd2*-knockdown embryos. Given the anterior/posterior asymmetries in cell shape and cilia clustering [27,30], we asked whether this was also reflected in CFTR-GFP distribution. Therefore, we compared the CFTR-GFP signal of the anterior half of the KV with that of the posterior KV region (Figure 3J,M). We observed no significant differences in non-injected and mismatch-MO control embryos (Figure 3J,K,M). However, the knockdown of PC2 led to a significant enrichment of CFTR-GFP signal in the anterior half of the KV (*pkd2*-augMO anterior-MFI = 1.4 × 10^6^ ± 0.6 × 10^6^ vs. *pkd2*-augMO posterior-MFI = 1.2 × 10^6^ ± 0.7 × 10^6^, with *p* = 0.0264) (Figure 3L,M). We reported several times before that the anterior KV region is where fluid flow presents consistently higher speed in WT embryos [29,30,34], thus perhaps flow shear stress can further induce CFTR recruitment.

### 2.3. Tolvaptan Effect

It is well established that ADPKD kidney tissues present higher vasopressin-induced intracellular cAMP levels, which contributes to the activation of CFTR [9]. Therefore, as we claim that KV closely mimics kidney cyst inflation, we cannot rule out that cAMP production may also have a role in KV inflation. We detected the expression of the *vasopressin receptor 2-like* gene in a KV-specific microarray analysis performed by our group (unpublished data), which was not affected by the knockdown of PC2 (Appendix A).

As above, we took the KV volume as a live-readout of CFTR activity [27] and tested the efficacy of tolvaptan, a specific V2R antagonist, in affecting the KV inflation process and in rescuing the CFTR-mediated KV enlargement caused by the knockdown of *pkd2*.

Here, we used the *Tg(sox17:GFP)s870* zebrafish line. Considering the middle focal plan and respective orthogonal views, *pkd2*-knockdown embryos presented KVs with 1.5 times the volume of their WT siblings (*pkd2*-knockdown KV_volume_ = 181 × 10^3^ ± 69 × 10^3^ μm^3^ vs. WT KV_volume_ = 120 × 10^3^ ± 56 × 10^3^ μm^3^, with *p* < 0.0001) (Figure 5A,B,J). WT and mismatch-MO KVs and the respective DMSO-treated controls were equivalent (Figure 5A,C,D,F,J). These new measurements confirmed our results previously obtained with *ras*:GFP transgenic embryos [27] and the results presented here with *TgBAC(cftr-GFP)pd1041* embryos (Appendix A). Additionally, we show, for the first time, that the KV enlargement characteristic of the *pkd2*-knockdown embryos is efficiently rescued by the simultaneous injection with the *pkd2*-augMO and 1000 ng of full-length *Xenopus pkd2-mRNA*, at the one-cell stage (*pkd2*-knockdown KV_volume_ = 181 × 10^3^ ± 69 × 10^3^ μm^3^ vs. rescue KV_volume_ = 89 × 10^3^ ± 45 × 10^3^ μm^3^, with *p* < 0.0001). These rescued embryos presented KV volumes even smaller than the non-injected controls (rescue KV_volume_ = 89 × 10^3^ ± 45 × 10^3^ μm^3^ vs. WT KV_volume_ = 120 × 10^3^ ± 56 × 10^3^ μm^3^, with *p* = 0.0379).

To demonstrate how useful the KV can be as a model organ for screening compounds that may prevent cyst enlargement through CFTR, we showed that by treating embryos with 100 µM tolvaptan, from 4 ss until 9–10 ss, caused a significant reduction of the KV luminal volume of WT embryos (100 µM tolvaptan-treated WT KV_volume_ = 64 × 10^3^ ± 29 × 10^3^ μm^3^, i.e., 0.5 times the size of the non-treated WT KVs, *p* < 0.0001) (Figure 5G,J). On the other hand, it also rescued the *pkd2*-knockdown KV volumes to values significantly lower than those of non-treated *pkd2*-knockdown embryos (100 µM tolvaptan-treated *pkd2*-knockdown KV_volume_ = 102 × 10^3^ ± 59 × 10^3^ μm^3^, i.e., 0.6 times the size of the non-treated *pkd2*-knockdown, with *p* = 0.0002) (Figure 5H,J).

In an attempt to facilitate the use of the KV in a screening context, we verified whether we could also assess its significant enlargement upon the *pkd2*-knockdown and its rescue by a pharmacological approach, just considering the area of the KV middle focal plan (Figure 5K). We observed that *pkd2*-knockdown embryos presented KVs with 1.3 times middle focal plan area of their WT siblings (*pkd2*-knockdown KV_middle focal plan area_ = 4.6 × 10^3^ ± 1.4 × 10^3^ μm^2^ vs. WT KV_middle focal plan area_ = 3.5 × 10^3^ ± 1.2 × 10^3^ μm^2^ with *p* = 0.0003). WT and mismatch-MO KVs and the respective DMSO-treated controls were equivalent. Moreover, this is significantly rescued either by co-injecting the *Xenopus pkd2-mRNA* or by 100 µM tolvaptan treatment. However, the observed differences were not as accentuated as those observed by measuring the KV volume, suggesting that its enlargement results from the sum of two factors—slices with larger areas and a higher number of slices, i.e., deeper KVs. This was, indeed, the case—KVs were bigger in all three dimensions. In fact, *pkd2*-knockdown KVs had on average 117 ± 19 slices, which was significantly more than WT KVs (97 ± 25 slices), *p* < 0.0001 (Figure 5L). Therefore, we concluded that, although useful for a faster screening, some information may have been lost if we looked only for the KV middle plan area when screening compounds that may prevent cyst enlargement through CFTR.

Interestingly, although tolvaptan reduced the KV volume of the *pkd2*-knockdown embryos compared to WT embryos, the exact same treatment, i.e., 100 µM tolvaptan from 4 to 9–10 ss, did not rescue their heart situs defects and curly-up tail phenotype (Figure 5M). Indeed, as previously described by our group [27] and by others [36], the knockdown of PC2 induced a curly-up tail phenotype and about 56% of right-sided or central hearts, which were not corrected by tolvaptan treatment (Figure 5M). This observation suggests that the laterality problems associated with the *pkd2* knockdown do not depend on the KV enlargement but rather on another mechanism downstream of PC2. And, indeed, we know, from our previous work, that in addition to the volume, the knockdown of PC2 affects other variables of the KV, such as cilia length, flow dynamics, and architecture [27,34]. Moreover, although significant, the reduction in KV volume induced by 100 µM tolvaptan treatment did not induce per se heart situs defects in WT embryos. Explaining this observation is the fact that tolvaptan-treated WT KVs had on average 2.2 × 10^3^ ± 0.6 × 10^3^ µm^2^ of middle focal plane area, which is above the size threshold (1.3 × 10^3^ µm^2^) defined by Gokey et al. for robust left–right patterning of zebrafish embryo [38]. Still, it will be interesting, in the future, to study the effect of tolvaptan over the entire KV time window (from 1 to 14 ss) to better understand whether V2R and cAMP signalling play a role in left–right patterning.

PKD1 mutations are associated with more severe ADPKD phenotypes. It would have been interesting to study the effect of the knockdown of *pkd1a* or *pkd1b* (the zebrafish orthologues of the human *PKD1*) on CFTR protein in the KV cells. However, this is not biologically feasible. Indeed, according to the in situ hybridization experiments of England et al., no *pkd1a* or *pkd1b* mRNA expression was detected in dorsal forerunner cells (DFCs) or in KV cells during the zebrafish development [39]. There is evidence that instead of *pkd1a*/*b*, *pkd1L1* is the relevant player in the KV. This paralogue has been demonstrated to be expressed on both DFCs and KV cells at its transcriptional level [39] and along KV cilia at the protein level [40]. Pkd1L1 is thus expected to be the partner of PC2 in setting up the correct left–right laterality of the internal organs of the zebrafish [39]. Supporting this, Pkd1L1 medaka morphants exhibit abnormal left–right patterning [41]. However, to our knowledge, defects in *pkd1L1* expression have never been linked to the formation of kidney cysts or ADPKD. Moreover, *pkd1L1* and its human orthologue *PKD1L1* have marked structural differences from their respective paralogues, *pkd1a*/*pkd1b* and *PKD1* [39], suggesting potential differences in the roles played by their encoded proteins. Therefore, although interesting for the left–right field, we do not expect the study of the impact of Pkd1L1 knockdown in zebrafish KV to be useful in the context of the ADPKD cyst inflation process. And, thus, the important discussion of how PC1 mutations impact on CFTR activity and cyst inflation must be further explored using other animal or cellular model systems.

The present work grants the potential of using the zebrafish KV as an organ system to study the link between the lack of PC2, the consequent changes in Ca^2+^ signalling, and CFTR expression/activation. Although not being a suitable model to study the physiopathology of ADPKD, namely the kidney dysfunction induced by the multiple fluid filled cysts present in human ADPKD kidneys, the KV mimics an isolated cyst. It is, therefore, a simple and low-cost model for screening molecules that prevent ADPKD cyst growth, in a preliminary phase of drug development.

We summarize our main findings and hypotheses in Figure 6. It is well established that the abnormal cyst inflation process in ADPKD is highly dependent on sustained CFTR activation [9,10,11,12,13,14]. This is, in turn, granted by the abnormally high intracellular levels of cAMP of ADPKD tissues [9]. Here we also show that, similarly to what happens in an ADPKD cyst, KV enlargement in *pkd2*-knockdown fish is also dependent on vasopressin signalling and can be prevented by tolvaptan. But, additionally, we demonstrated here that *pkd2*-knockdown KV cells presented significantly higher levels of CFTR-GFP in their apical region, compared to their respective controls. This data brings novel alternatives for CFTR-mediated ADPKD cyst inflation, which may rely on enhanced expression levels of CFTR at the apical membrane of the cyst-lining cells.

## 3. Materials and Methods

### 3.1. Fish Strains

The two following zebrafish lines, both of AB background, were used for this work: a KV reporter line *Tg(sox17:GFP)s870* [42] and a CFTR-GFP transgenic line, the *TgBAC(cftr-GFP)pd1041* [28]. Zebrafish adults were maintained at 28 °C. Embryos obtained from incrosses were incubated in E3 medium at 25 °C or 28 °C and staged as described elsewhere [43].

### 3.2. Morpholino Microinjections

The knockdown of PC2 was induced using the previously reported *pkd2*-augMO [27,35,36]. We injected 1.8 ng into one-cell stage zebrafish embryos and 1.8 ng of a mismatchMO was used as control. Both morpholinos were purchased from Gene Tools LLC (Philomath, OR, USA). Additionally, 1000 pg of *Xenopus pkd2* mRNA was injected at the one-cell stage in combination with *pkd2*-augMO, to try to rescue the KV enlargement caused by the latter.

### 3.3. Pharmacological Treatments

Stock solution: 10 mM tolvaptan (Sigma-Aldrich, St. Louis, MO, USA) in DMSO (Sigma-Aldrich). Embryos were treated with 100 µM tolvaptan in E3 from 4 ss onwards and imaged at the 9–10 ss.

### 3.4. Heart Laterality Scoring

The heart jogging was evaluated at 30 hpf by observing the embryos from their ventral side, using a stereoscope (SMZ745, Nikon Corporation, Tokyo, Japan).

### 3.5. Immunofluorescence on Whole-Mount Embryos

The immunofluorescence on whole-mount embryos analysis was performed as previously described [27]. Dechorionated 10–11 ss embryos were fixed in 80:20 (*v*/*v*) methanol:DMSO for 1 min and rehydrated in sequential 5 min incubations in crescent dilutions of methanol in PBS. After permeabilization and blocking, embryos were incubated overnight at 4 °C with 1:200 diluted anti-PC2 polyclonal antibody (GTX113802, GeneTex, Irvine, CA, USA) and, subsequently, with 1:400 diluted anti-acetylated α-tubulin monoclonal antibody (T7451, Sigma-Aldrich); 1:500 diluted Alexa Fluor 488 or 546 conjugated secondary antibodies (Molecular Probes, Eugene, OR, USA) were used. Flat-mounted embryos were analyzed with confocal fluorescent microscopy (Zeiss LSM710, Zeiss, Oberkochen, Germany) and their whole KVs were scanned with *z*-sections of 0.5 μm. *Z*-stack series were analyzed using ImageJ software (version 1.53a, National Institutes of Health, USA) and were used to quantify the % of KV cilia positively stained for PC2.

### 3.6. Live-Imaging and KV Volume Determination

The live-imaging and KV volume determination was performed as previously described [27]. *Tg(sox17:GFP)s870* and *TgBAC(cftr-GFP)pd1041* embryos were mounted in a 2% (*w*/*v*) agarose mold and covered with E3 medium. For volume evaluation, the whole KVs of 9–10 ss embryos were scanned by confocal-live microscopy, with *z*-sections of 0.5 μm and acquisition rate lower than one frame per second. Movies were then analyzed in ImageJ for volume estimation. Using the ImageJ plugin Measure Stack, the KV was delineated and its area was measured in all focal planes. The volume resulted from the sum of the measurements of all focal planes.

### 3.7. Evaluation of CFTR-GFP Mean Fluorescence Intensity

CFTR-GFP MFI was evaluated using the 2D image of the whole KV *TgBAC(cftr-GFP)pd1041* embryos. This was obtained from the sum of all slices of the KV stack obtained by live confocal microscopy.

For CFTR-GFP MFI was evaluated in (1) the whole KV, (2) in the apical region of the cells facing the lumen of the organ, and (3) in the anterior versus posterior halves of the KV. For the first, the full area of the image (2.0 × 10^4^ µm^2^) was considered and this was equal for all the analyzed samples. The second measurement was performed by defining a ring-shaped area, limited by the apical membrane and the sub-apical region of KV cells. The third was obtained by dividing the full area of the image into two halves, anterior (upper part of the image) and posterior (the lower part). For all these measurements, the following parameters were considered: mean gray value, i.e., the sum of the gray values of all pixels in the selected area divided by the total number of pixels; and integrated density, i.e., the product of the selected area and the mean gray value. The normalized MFI was determined by normalizing the integrated density with the background MFI, using the following equation: normalized MFI = integrated density of selected area—(selected area × background mean gray value).

### 3.8. Flow Cytometry Analysis

For each replicate, about 200 *TgBAC(cftr-GFP)pd1041* embryos were dechorionated at 10 ss, with pronase (2 mg/mL) (EMD Millipore, MERK, Germany) and washed extensively in Danieux buffer. Cells were dissociated by manual pipetting in DMEM-F12 supplemented with 5 mM EDTA (Sigma-Aldrich). Cells were centrifuged at 700 × *g* and re-suspended in 1 mL of the same medium (step performed three times). Once dissociated, embryonic cells were re-suspended in only 300 µL of PBS and were acquired directly to the flow cytometer BD FACS-Canto II (BD Biosciences). After excluding debris and medium component cell agglomerates and auto-fluorescent cells, using AB control, we determined the MFI of the CFTR-GFP-positive single cells. For each experiment, fluorescence intensity values were normalized with the auto-fluorescence of the *TgBAC(cftrGFP)pd1041* embryos. Both mean and median of the fluorescence intensity in the GFP (530/30 nm BP) channel, the number of GFP-positive cells and the percentage of those in the entire population of cells (after excluding debris), were calculated by the acquisition software BD FACS-DIVA^™^ (version 8.0.1) (BD Biosciences). The presented flow cytometry plots were generated with the FlowJo^®^ software (version 10.3) (FlowJo LLC, BD Biosciences).

### 3.9. Statistics

Statistical analysis was performed with Prism 6 (Graphpad, La Jolla, CA, USA). Samples were tested for normality with the Shapiro–Wilk test or the Kolmogorov–Smirnov test and for equal variances with the F-test. Differences between samples having a normal distribution and equal variances were analyzed for statistical significance using the *t*-test (two-tailed). Normal samples having significantly different variances were statistically compared with the *t*-test (two-tailed) with the Welch correction. Additionally, samples that did not passed normality tests were analyzed using the Mann–Whitney test (two-tailed). The paired *t*-test (two-tailed) was used to compare the anterior versus posterior MFIs for each analyzed situation. Statistically significance was considered when *p* < 0.05. Results are expressed as averages ± SD of *n* observations.

## 4. Conclusions

In conclusion, the CFTR-mediated KV inflation depends on the sustained activation of CFTR by cAMP, levels of which are, at least partially, controlled by vasopressin-signaling. This fact, added to the enhanced membrane stability of CFTR observed upon the reduction of PC2, accounts for the KV enlargement observed in the *pkd2*-knockdown embryos. Based on our results we consider that this zebrafish organ system offers great potential for screening molecules that interfere with CFTR trafficking, stability and channel activity in preventing ADPKD cyst growth.

## Figures and Tables

**Figure 1 ijms-22-09013-f001:**
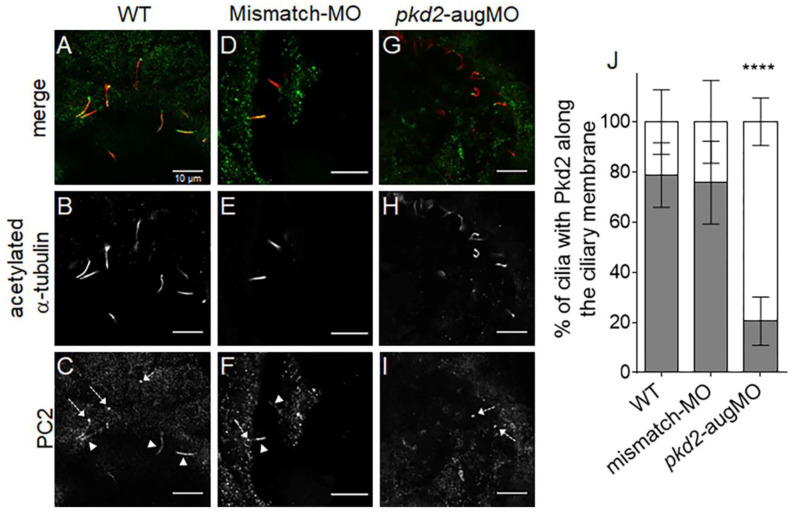
PC2 staining along the KV cilia from 10 ss embryos. (**A**–**I**) Confocal images showing immunolocalization of PC2 in KV cells at the 10–11 ss in WT (**A**–**C**), mismatch-MO (**D**–**F**), and PC2-knockown (*pkd2*-augMO) (**G**–**I**) embryos. White arrowheads indicate PC2 detected along cilia and dashed arrows indicate PC2 at the cilia basal body. (**J**) Quantification of the percentage of cilia having PC2 signal along their membrane (gray bars) versus those with no PC2 staining along their membrane (white bars) in WT (*n* = 16), mismatch-MO (*n* = 8), and PC2-knockown (*n* = 8) embryos, immunodetected for acetylated α-tubulin. All samples followed a normal distribution and presented equal variances. Differences were, thus, statistically tested by *t*-test, **** *p* < 0.0001. Scale bars: 10 μm.

**Figure 2 ijms-22-09013-f002:**
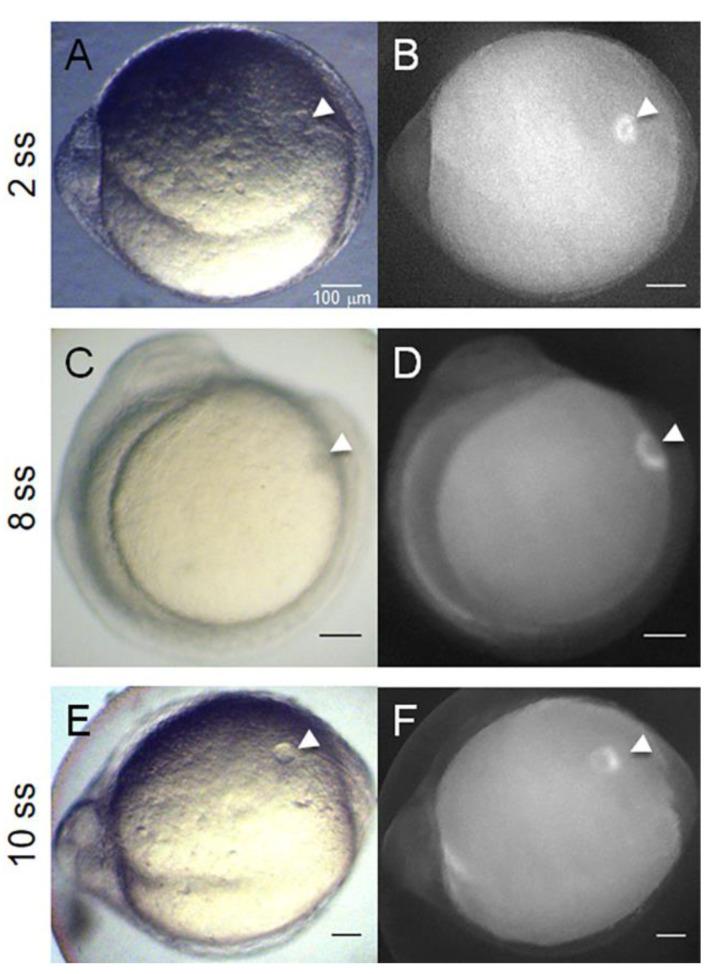
The zebrafish KV—a model organ to study cyst inflation. The KV localized at the tail region of a 2–14 ss zebrafish embryo. *TgBAC(cftr-GFP)pd1041* zebrafish line characterization. (**A**,**C**,**E**) are bright field captured images. (**B**,**D**,**F**) were acquired by fluorescence stereomicroscopy. White arrowheads indicate the KV. Scale bars: 100 μm.

**Figure 3 ijms-22-09013-f003:**
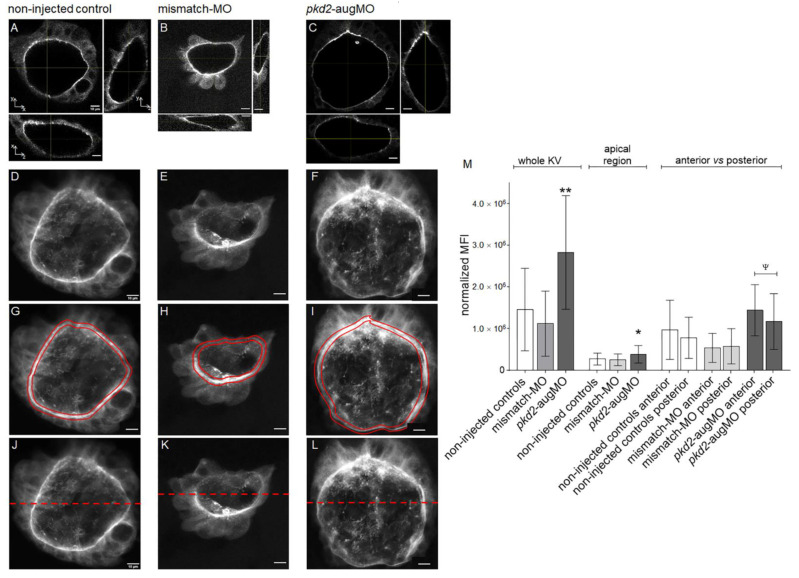
Estimated CFTR-GFP protein level in KVs of *TgBAC(cftr-GFP)pd1041* embryos. (**A**–**L**) Confocal live-scan analysis. The middle plan along the xy axis and its respective orthogonal views (along xz and yz axes) and the image resulting from the sum of all slices of the confocal live-microscopy scan of the whole KV are, respectively, shown for the most representative non-injected control (**A**,**D**), mismatch-MO (**B**,**E**), and *pkd2*-knockdown (**C**,**F**) embryos. (**G**–**I**) The apical region of the KV cells of (**D**,**E**,**F**) is highlighted in red. (**J**–**L**) anterior (upper) versus posterior (lower) parts of the KV are highlighted. (**M**) MFI determined for the whole KV, KV cells’ apical region and KV’s anterior versus posterior regions of non-injected control (*n* = 15), mismatch-MO (*n* = 6), and *pkd2*-knockdown (*n* = 20) embryos. Median averages ± SD are indicated. *t*-test was used to compare whole KV MFIs of *pkd2*-augMO versus non-injected embryos (samples with normal distribution and equal variances), ** *p* < 0.01. Mann–Whitney test was used to compare the MFIs of KV cells’ apical region of *pkd2*-augMO versus non-injected embryos (samples that did not passed normality tests), * *p* < 0.05. Paired *t*-test was used to compare the anterior versus posterior MFIs for each situation (all samples having normal distribution), ^ψ^ *p* < 0.05. Scale bars: 10 μm.

**Figure 4 ijms-22-09013-f004:**
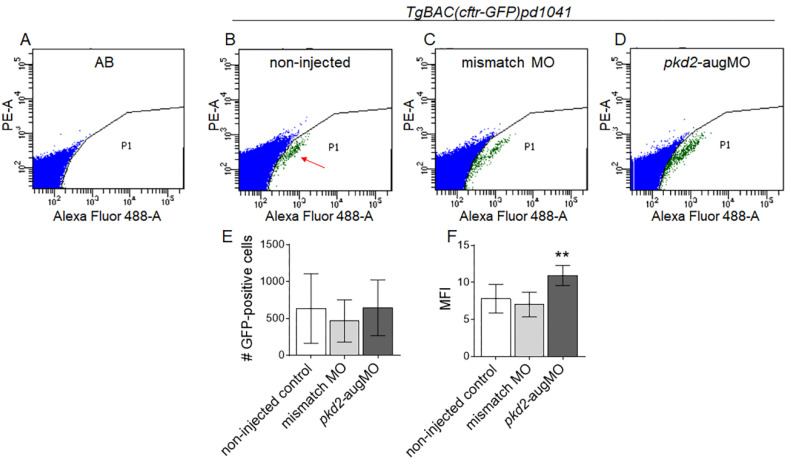
Flow cytometry analysis of the CFTR-GFP protein level in *TgBAC(cftr-GFP)pd1041* embryos. Flow cytometry plots representative of AB control (**A**) and non-injected *TgBAC(cftr-GFP)pd1041* 8–10 ss (**B**) embryos, for the established limiting gates. Red arrow indicates the GFP-positive cell population. (**C**,**D**) Flow cytometry plots representative of the GFP-positive cells of mismatch-MO (**C**) and *pkd2*-augMO (**D**) embryos. (**E**) Number of analyzed GFP-positive cells of non-injected controls (11 replicates), mismatch-MO (3 replicates), and *pkd2*-knockdown embryos (5 replicates). Each replicate had about 200 embryos. (**F**) MFI determined for GFP-positive cells of non-injected controls, mismatch-MO, and *pkd2*-knockdown embryos. Median averages ± SD are indicated. As the *pkd2*-augMO injected embryos sample did not follow a normal distribution, the Mann–Whitney test was used to compare *pkd2*-augMO versus non-injected embryos MFIs, ** *p* < 0.01.

**Figure 5 ijms-22-09013-f005:**
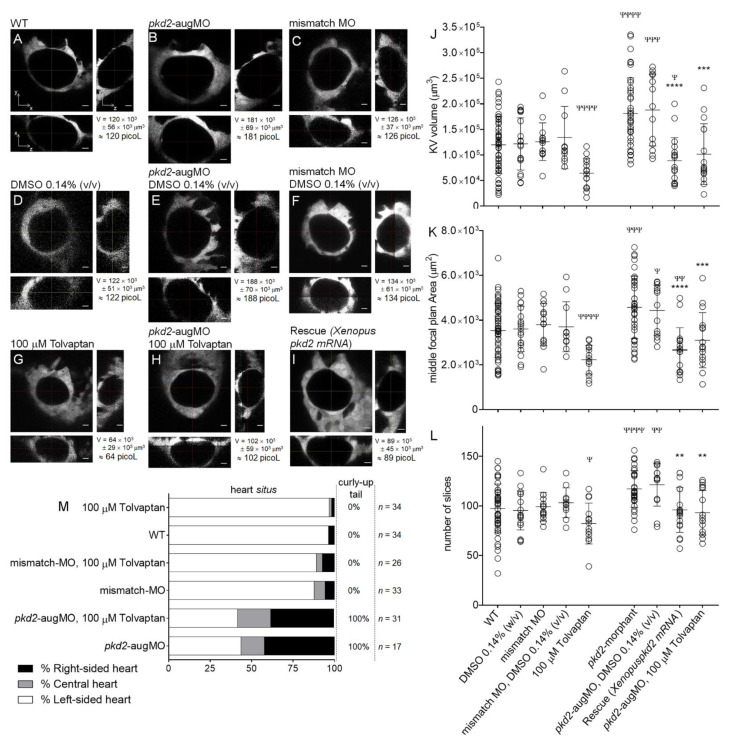
KV volume, a live readout of the CFTR activity. (**A**–**I**) Confocal live-microscopy scans of whole KVs from *sox17*:GFP transgenic embryos, at 10–11 ss. The middle focal plane along the xy axis and its respective orthogonal views (along xz and yz axes) are shown for the most representative embryos: WT (**A**), *pkd2*-knockdown (**B**), mismatch MO (**C**), WT + 0.14% (*v*/*v*) DMSO (**D**), *pkd2*-knockdown + 0.14% (*v*/*v*) DMSO (**E**), mismatch MO + 0.14% (*v*/*v*) DMSO (**F**), WT + 100 μM tolvaptan (**G**), *pkd2*-knockdown + 100 μM tolvaptan (**H**), and rescue (*pkd2*-knockdown + *Xenopus pkd2-mRNA*) (**I**). KV volume is indicated in μm^3^ and in picoliters. Scale bars: 10 μm. (**J**,**K**,**L**) Estimated KV volumes (μm^3^), middle focal plan area (μm^2^) and number of slices of the z-stacks for WT (*n* = 51), WT treated with 0.14% (*v*/*v*) DMSO (*n* = 19), mismatch-MO (*n* = 14), mismatch MO + 0.14% (*v*/*v*) DMSO (*n* = 11), 100 μM tolvaptan (*n* = 14), *pkd2*-knockdown (*n* = 38), *pkd2*-knockdown treated with 0.14% (*v*/*v*) DMSO (*n* = 13), 100 μM tolvaptan (*n* = 16), and rescue (*pkd2*-knockdown + *Xenopus pkd2-mRNA*) (*n* = 18). Mean ± S.D. All samples, except mismatch MO + 0.14% (*v*/*v*) DMSO volume and middle focal plan area, *pkd2*-knockdown + 100 μM tolvaptan volume, *pkd2*-knockdown number of slices, and *pkd2*-knockdown + 0.14% (*v*/*v*) DMSO number of slices, had a normal distribution. The comparisons of samples that did not have normal distribution with respective WT and *pkd2*-knockdown samples were analyzed with the Mann–Whitney test. *t*-test with the Welch correction was used to compare WT + 100 μM tolvaptan with WT volume and middle focal plan area (samples with unequal variances). All other comparisons to WT and to *pkd2*-knockdown samples were made using the *t*-test. ^ψ^
*p* < 0.05, ^ψψ^
*p* < 0.01, ^ψψψ^
*p* < 0.001, and ^ψψψψ^
*p* < 0.0001, significantly different from WT; and **** *p* < 0.0001, *** *p* < 0.001, and ** *p* < 0.01, significantly different from *pkd2*-knockdown embryos. (**M**) Heart position defects and curly-up tail phenotype of WT and mismatch-MO controls and PC2 knockdown embryos, treated with 100 μM tolvaptan. *n*, number of scored embryos.

**Figure 6 ijms-22-09013-f006:**
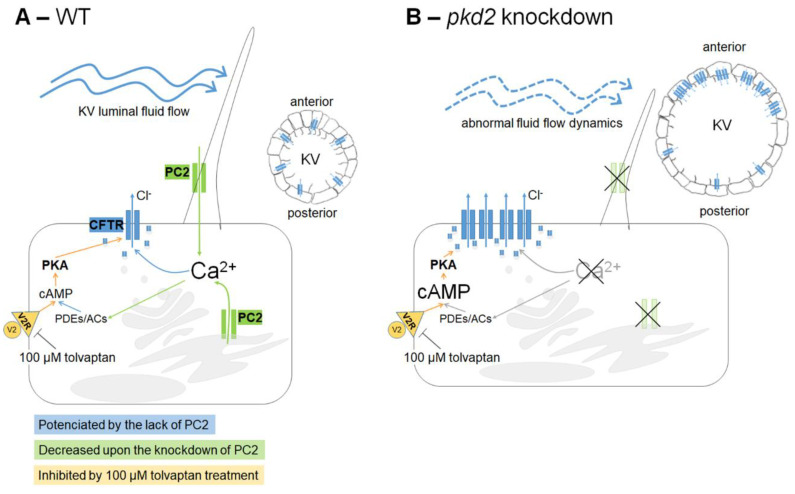
Working model of the cross-talk between PC2 and CFTR in the KV inflation. (**A**) As in kidney cysts, we extrapolate that in WT KV epithelial cells, once stimulated by the luminal fluid-flow, PC2 raises intracellular Ca^2+^ transients. These should maintain the basal intracellular levels of cAMP, through regulation of Ca^2+^-dependent adenylyl cyclases (ACs) and phosphodiesterases (PDEs). cAMP is required for the normal rate of CFTR activity. This ensures the CFTR-mediated transport of Cl^−^ and subsequent movement of water into the KV lumen, allowing its inflation. Additionally, cAMP levels are also dependent on V2R since we know that its pharmacological inhibition by 100 µM tolvaptan impairs the KV inflation. (**B**) The knockdown of *pkd2* changes several variables of the KV: volume enlargement [27], different cell shapes [27] and, therefore, different architectures with the consequent loss of the anterior cilia cluster [27,34], shorter cilia [34], and weakened and homogeneous luminal flow [34]. The lack of PC2 should lower the intracellular levels of Ca^2+^, which, through still unknown mechanisms, leads to a significant increase of CFTR expression at the apical region of the cell (membrane and subapical zone), namely in the anterior part of the KV. This justifies the CFTR-mediated KV enlargement observed upon lack of PC2. It is still unknown if the resulting raise in the cAMP levels, induced by the low Ca^2+^ levels, further potentiates the CFTR activity. However, mimicking a kidney cyst, pharmacological inhibition of KV V2R by 100 µM tolvaptan was enough to rescue the KV enlargement of these embryos.

## Data Availability

The raw data supporting the conclusions of this article will be made available by the authors, upon request, without undue reservation.

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
