# Peer review of "Zebrafish Model as a Screen to Prevent Cyst Inflation in Autosomal Dominant Polycystic Kidney Disease"

_ijms, 2021, doi:10.3390/ijms22169013_

Round 1

Reviewer 1 Report

In this manuscript

  • Authors demonstrate that knockout of the pkd2gene (injection of the anti-pkd2mRNA blocking morpholino) causes reduced expression of PC2 in the KV cilia.
  • Authors demonstrate that knockdown of PC2 leads to the apical accumulation of CFTR protein. To accomplish this authors had to overcome the absence of immunostaining with commercial available human CFTR antibodies.
  • Authors demonstrate increased volume of the KV vesicles as the result of increased expression of CFTR protein.
  • Authors tested the effect of tolvaptan, the only drug approved for human use in ADPKD patients, using KV volume as a surrogate for CFTR activity

Main concerns

In my opinion the manuscript addresses an important aspect in the search for a consistent treatment for ADPKD. Authors proposed a simple and considerably less expensive model for testing drugs in a preliminary phase of drug development. The only concern is how this model reproduces what happens in humans as Kupfer Vesicle is a strange organ that keep only a few resemblance to the kidneys as it is a single organ in contrast with multiple fulfilled cysts in human PKD.

Minor concern

  • As the number of experiments is less than 30 and the distribution of the variable is not normal (at least normality was not tested), I suggest to use a non-parametric statistical analysis.
  • Conceptually, as the KV develops at age ss 1 to 14, it would be interesting to ascertain if tolvaptan would be effective in earlier or later stages of development.

Reviewer 2 Report

Oliveira and colleagues postulate the embryonic zebrafish as a model for screening to prevent cyst inflation in Autosomal Dominant Polycystic Kidney Disease. In particular, they focus on the transient zebrafish Kupffer’s Vesicle (KV) as model system. They show that PC2 knockdown causes a reduction in the number of KV cilia expressing PC2. PC2 knockdown leads to accumulation of CFTR-GFP at the apical region of the KV cells. KV enlargement is prevented by injection of Xenopus pkd2 mRNA and by 100 μM tolvaptan treatment.

The PC2 knockdown model has been published already several years ago.

Also, the role of cftr in KV cells has been documented already several years ago.

The complete signalling cascades mentioned within the current manuscript were described for several model organisms already.

The use of Students’ or Mann-Whitney tests for non-injected, mismatch or PC2 knockdown groups is not appropriate.

The effects of the PC2 knockdown on CFTR-GFP expression are relatively small and only detectable in the MFI levels, which is especially difficult to interpret from the Figure 4 data. As neither Western Blot nor immunofluorescence worked for CFTR detection, the experiments largely depend on the assumption that the GFP behaves similar to CFTR protein.

The focal plans represented in Figure S2 are not comparable, not suitable to create correct volume measurements.

The 100 μM tolvaptan treatment part is pharmacologically irrelevant given the dose used.

The injection experiments with Xenopus pkd2 mRNA give the expected result in terms of KV volumes, but it is unclear whether this corresponds to normalized pkd2 protein levels or an overexpression model.

Line 88: “ADPKD mouse models (12, 26), although useful, are limited for live-imaging experiments” is not accurate. High resolution ultrasound can be used in the mouse models. Only the imaging approach is different from zebrafish embryos.

Reviewer 3 Report

This is a nice paper. However, I have some comments.
The findings from this paper are excellent and worthy to review.
This manuscript contained some questions described below.
I think this paper is interesting, this review contributes to future's clinical

medicine largely.

I have some questions from a point of view of clinical medicine.

I have a clinical impression that polycystic kidney disease progresses differently depending on the size of the cyst. Is there any effect of tolvaptan or renal prognosis between cystic kidney disease with large cysts and cystic kidney disease with small cysts? If the two have different prognosis and therapeutic effects, please tell us if there is a difference in the degree of CFTR activity in this experiment.

The theme of this experiment is the case of PC2 abnormality, but please tell us what kind of result can be expected in the case of PC1 abnormality.

Reviewer 4 Report

The authors presented a well-designed kupffer's vesicle as an in vivo model for potential therapeutic compounds to treat ADPKD. The study is well conducted and provides a potential new target for the treatment of ADPKD.

Author Response

We acknowledge the reviewer for nicely pointing out the strengths of our study.

Round 2

Reviewer 2 Report

In the response, the authors mainly agree with the reviewers comments and only defend the limitations of their study. Only limited improvement is provided especially in terms of lack of novelty and inappropriate statistics.